# Developing and validating a HEalthCare NAvigation Competency (HECNAC) Scale for refugees in the United States

Sarah Yeo[1]*, Inseok Lee[2], John Ehiri[3], Priscilla Magrath[3], Kacey Ernst[4], Yu Ri Kim[5], Halimatou Alaofè[3]

1 The University of Arizona Cancer Center, Tucson, Arizona, United States of America, 2 Institute for Global Health, University College London, London, United Kingdom, 3 Health Promotion Sciences Department, Mel and Enid Zuckerman College of Public Health, University of Arizona, Tucson, Arizona, United States of America, 4 Epidemiology and Biostatistics Department, Mel and Enid Zuckerman College of Public Health, University of Arizona, Tucson, Arizona, United States of America, 5 Asia-Pacific Research Center & School of International Studies, Hanyang University, Seoul, Republic of Korea

* syeo@arizona.edu

**Data Availability Statement:** All relevant data are within the manuscript and its Supporting Information files.

## Abstract

The complex healthcare system in the United States (US) poses significant challenges for people, particularly minorities such as refugees. Refugees often encounter additional layers of challenges to healthcare navigation due to unfamiliarity with the system, limited health literacy, and language barriers. Despite their challenges, it is difficult to identify the gaps as few tools exist to measure navigation competency among this population and many conventional tools assume English proficiency, making them inadequate for refugees and other immigrants. To address this gap, this study developed and validated a HEalthCare NAvigation Competency (HECNAC) Scale tailored to refugees' needs. The scale development process followed three phases: domain identification through a literature review and stakeholder interviews (n = 15), content validation through the Delphi method (2 rounds, n = 12), and face validity assessment via cognitive interviews (2 rounds, n = 4). Based on a literature review and stakeholder interviews, the initial version of the scale was developed, including ten domains and 47 items. An introductory email concerning the scale and the Delphi process was subsequently sent to 21 eligible experts, including staff from refugee resettlement agencies, health care providers serving refugee communities, and refugees. Twelve experts completed the two rounds of the Delphi, resulting in a consensus on 39 items. After conducting cognitive interviews with 4 Afghan refugees, the scale was finalized with ten domains and 35 items. The finalized scale captures multifaceted aspects of healthcare navigation crucial for refugees, organized into domains such as health system knowledge, insurance, making an appointment, transportation, preparing for a visit, in the clinic, interpretation, medicine, medical bills, and preventive care. Overall, the HECNAC Scale represents a significant step towards understanding and assessing refugees' competencies in navigating the US healthcare system. It has the potential to guide tailored interventions and standardized training curricula and ultimately mitigate persistent barriers faced by refugees in accessing healthcare services.

**Funding:** This research is supported in part by NIH T32 CA078447 and the funders had no role in study design, data collection and analysis, decision to publish, or preparation of the manuscript.

**Competing interests:** The authors have declared that no competing interests exist.

## Introduction

The healthcare system in the United States (US) is notoriously complex and fragmented, posing significant challenges even for individuals who are native to the country [1]. With complex processes, constantly changing rules and regulations, and a complicated insurance system, understanding how to access and navigate the healthcare system and services effectively can be challenging [1]. This difficulty can be exacerbated for minorities, such as refugees, who often have limited access to resources and information and encounter additional layers of obstacles.

Refugees face challenges to healthcare navigation due to unfamiliarity with the system and lack of health literacy and proficiency in the English language, even years after their initial arrival [2,3]. In their countries of origin, refugees are often accustomed to a healthcare system that operates differently. For example, there may be no need for medical appointments, a different insurance system, and pharmacies located near or within hospitals.

Since the Refugee Act was enacted in 1980, the US has resettled over 3.2 million refugees from more than a hundred different countries. Upon initial resettlement in the US, refugee resettlement agencies provide training to assist refugees in adjusting to their new environment and health system. Nevertheless, there is a lack of consistency in the training offered by these organizations, leading to disparities in the depth and quality of training delivered by various organizations and caseworkers. Additionally, the majority of programs targeting refugees are short-term [4] and do not consider whether refugees have acquired sufficient skills to navigate the complicated healthcare system in the country. As a result, refugees are left to navigate on their own or must rely on informal support from family or friends once the time-limited support ends [5]. This can be particularly difficult for refugees with limited education, poor English proficiency, and minimal social networks.

In response to barriers to healthcare access, health navigation emerged as a promising approach for improving the healthcare journey and health outcomes among minorities. The concept of health navigation can be traced back to the American Cancer Society National Hearings on Cancer in the Poor in 1989, which revealed various barriers to timely cancer screening, diagnosis, treatment, and supportive care. These included financial, communication and informational, medical system, and emotional barriers [6]. Responding to these challenges, the first patient navigation program was initiated in Harlem, New York, in 1990 as a measure to address these challenges among black women. Since then, health navigation has evolved as a strategy to enhance health outcomes among marginalized communities by removing barriers that hinder timely diagnosis and treatment of cancer and other illnesses [6]. According to the existing body of literature, health navigation can be defined as the process of finding, accessing, and utilizing healthcare services effectively to achieve optimal health outcomes, and health navigation competency refers to the knowledge and skills required to achieve this goal [1,7,8].

While patient navigation programs are widely implemented and definitions of health navigation are well-established, few tools exist to measure navigation competency [7]. Additionally, existing health competency concepts are predominantly based on the idea of health literacy, and many assessment tools designed to measure health literacy assume English language proficiency. For instance, the Test of Functional Health Literacy in Adults (TOFHLA), one of the most widely used tools to evaluate adult health literacy, assesses health literacy in individuals who possess reading and writing proficiency in English [9]. The TOFHLA measures a patient's ability to comprehend written passages (Reading Comprehension) and numerical information (Numeracy) using actual healthcare-related materials. Consequently, these instruments are often inadequate for individuals who speak English as a second language or have limited proficiency in English, such as resettled refugees or any other groups of immigrants [9].

Furthermore, refugees face the obstacle of transitioning to an unfamiliar environment in which they must learn about and manage numerous aspects, such as using public transportation and knowing different levels of the healthcare system in the US. As a result, healthcare navigation competency among refugees extends beyond simply comprehending health-related information, and conventional health literacy or competency tools fail to capture the unique circumstances of refugees.

To address this issue, this study was conducted to identify essential competencies required for refugees to navigate the complex US healthcare system and develop a HEalthCare NAvigation Competency (HECNAC) Scale to assess the level of competencies among refugees. This tool can enable refugee-serving organizations to tailor their training to enhance refugees' competencies and provide more targeted support for those with limited resources and skills. One thing to note is that the scale was translated and tested only in Dari, the predominant language spoken in Afghanistan due to logistical and resource constraints. This decision was based on two key factors. Firstly, the greater need arose due to a significant increase in Afghan refugees in the US after the Taliban's takeover in 2019, as community partners highlighted the necessity of aiding their navigation of the healthcare system. Secondly, the first author's previous work and established connections with the population provided easy access to them.

## Methods

The study has been reviewed and approved by the Human Subjects Protection Program at the University of Arizona (IRB 2104716241). All the study participants provided informed written consent prior to participation. The scale development process in this study followed three phases based on academic literature [10,11]. The initial step involved identifying the domains and items of the scale through a review of relevant literature and stakeholder interviews. In the second phase, the study employed the Delphi method to evaluate the content validity of the scale. In the third phase, the scale was evaluated with a target population for face validity through cognitive interviews (**Fig 1**).

### Development of the HEalthCare NAvigation Competency (HECNAC) Scale

Several frameworks and models outline the processes by which patients access and navigate the healthcare system [12–14]. Based on the literature review of the frameworks, domains and items for the scale were first identified. Additionally, data were collected from multiple sources, including orientation curriculums for refugees and resources and guidelines for health care providers and community partners [10,11,15]. Interviews were conducted with fifteen stakeholders including health care providers (n = 4), cultural/clinical health navigators working with refugees (n = 4), staff from refugee resettlement agencies (n = 2) and governmental agencies (n = 2), and researchers studying refugee health (n = 3). Participants with over three years of experience working with refugees were intentionally chosen based on their occupational roles to gather diverse perspectives on healthcare navigation among refugees. Recruitment methods included referrals from community partners and snowball sampling. The stakeholders' insights and opinions were sought regarding the essential competencies that refugees need to possess in order to effectively navigate the healthcare system in the US. Through thematic analysis of the qualitative data of these interviews, additional domains and items were identified and added. Further details about the stakeholder interview are available elsewhere [16].

Except for the healthcare system domain, where respondents are prompted to indicate activities in certain situations, all other responses are rated on a 5-point Likert scale, ranging from Strongly Disagree to Strongly Agree. This decision was made based on the literature [11],

## Phase 1. Domain Identification and Item Generation

| Literature review and assessment of existing scales | Stakeholder interviews (n=15) |

## Phase 2. Content Validity through the Delphi

Invitation for participation was sent to 21 eligible experts and 14 participated in evaluating 47 items

1st Round (n=14): consensus reached for 39 items
The results sent back to the expert panel

2nd Round (n=12): consensus reached for 39 items

## Phase 3. Face Validity through Cognitive Interviews

Translation and back-translation

Cognitive interview (2 rounds, n=4 in total)

**Fig 1. Healthcare Navigation Competency Scale development process (informed by Boateng)** [10].

which recommends using five to seven categories for raters to ensure the reliability of the scale. Additionally, the literature suggests that an odd number of categories for bipolar scales, such as Strongly Agree to Strongly Disagree, allows raters the opinion to express neutrality when they are not sure [11]. Thus, neutral responses, Neither Agree nor Disagree, were included in the scale so that respondents could choose the option when they were unsure. After the domains and items of the scale were initially formulated, a panel of three reviewers assessed the preliminary draft of the scale. The review panel was comprised of a psychometric expert, a bilingual refugee woman proficient in English and her native languages, and a refugee resettlement staff member with over ten years of experience in refugee health. They provided feedback and comments to refine the scale.

### Content validation through the Delphi method

Content validity, which measures whether a scale adequately assesses the domain of interest [17], was measured through the Delphi method. Since its inception in the 1950s at the RAND Corporation, the Delphi method has been employed to achieve consensus among a group of experts [18,19]. It has been used to aggregate different ideas, predict uncertain issues, collect expert opinions, develop a framework, and reach consensus [20,21]. It is also helpful for assessing content validity through expert judges. In this study, the Delphi method was used to measure the content validity of the scale in accordance with existing literature [10]. While there are numerous variations of the Delphi, there are key components: the participation of a group of experts who provide input on a specific issue, an iterative process consisting of several rounds,

the avoidance of direct contact among experts to ensure anonymity, and the design of subsequent rounds informed by a summary of the previous rounds [18,22].

**The expert panel.**   The panel of experts was identified through recommendations from community partners, other researchers, and practitioners in the field. The Delphi panel included staff at refugee resettlement agencies, health care providers who serve refugee communities, cultural health navigators, and refugees. The experts, apart from refugees, possessed a minimum of three years of experience working with refugee populations.

**Number of experts.**   There is no consensus on the specific number of experts required for a Delphi study. The range can vary from a few to hundreds or a thousand [23]. According to a review of systematic reviews of Delphi studies in health sciences conducted by Niederberger and Spranger, the number of experts involved in Delphi studies usually falls within "the low to medium double-digit range [21]." In another study, it was suggested that a Delphi panel should consist of 15–30 participants within the same field, or 5–10 individuals per category from diverse professional groups. However, it was also noted that including more than 30 experts may not enhance the results [24]. In another study, which delineated the best practices for developing and validating scales, the authors recommended 5–7 expert judges [10]. Following the suggestions from the literature, twenty-one experts were reached and 14 experts participated in the Delphi.

**Consensus.**   Although there is no universally accepted standard for consensus, it is advisable to establish a clear consensus definition beforehand in order to ensure transparency [21–24]. This Delphi study employed the content validity ratio (CVR), which is suggested as a more advanced technique for content validation [11]. During the Delphi study, each item was assessed by the expert panel using a 4-point scale, with a score of 1 indicating high relevance and a score of 4 indicating irrelevance. The CVR for each item was then computed. Lawshe suggested a minimum CVR value of 0.99 for five or six raters, 0.85 for eight raters, and 0.62 for 10 raters [11]. As this study involved 14 expert raters, a minimum value of 0.62 was used to be on the conservative side. This value was set a priori to ensure transparency [21–24]. Any items that fell below these thresholds were removed.

$$CVR = \frac{n_e - \frac{N}{2}}{\frac{N}{2}}$$

Content validity ratio, CVR; $n_e$ = the number of panel members indicating an item to be essential (a rating of 3 or 4), N = the total number of panel members.

**The number of rounds.**   The most common number of rounds in the Delphi process is two or three rounds [21,25]. In this study, two rounds of the Delphi were conducted.

**Data collection.**   An invitation email, which includes an introduction to the study, the Delphi method, processes, and consent, was sent to 21 potentially eligible experts. They were asked to anonymously review and rate their level of agreement with each statement using a 4-point Likert scale. Additionally, they could provide opinions on items to be modified and suggestions for clarity and readability for each item. Their responses were collected through Qualtrics, and the response rate for each round was recorded to ensure the rigor of the technique [23]. For the second round, a revised version of the scale was created based on the agreement rate and anonymous feedback and was sent to the panel [21]. A summary of the results, including minimums, maximums, means, and CVR, was also provided. The feedback and comments from the panel were used to refine and modify the scale. The first round of surveys was conducted from June to July 2023, and the results were sent back to the expert panel in late July 2023. The second round of the survey was completed in early September 2023. The

reporting of the results follows the guidance on conducting and reporting Delphi studies (CREDES) [22].

## Translation and cognitive interviews for evaluating face validity and pretesting

When translating a tool into a different language, it is important to establish equivalence between the original version and the translated version [11]. In this study, equivalence was achieved through back-translation and cognitive interviews [26]. Before the cognitive interview, the finalized version was sent to a team of professional translators proficient in both English and Dari. After translation, another professional translator, a medical doctor in Afghanistan living in the US who had not seen the original English version, translated the Dari version back into English. The newly translated English version was then compared to the original version of the scale. A meeting was convened with the translators to resolve identified discrepancies, and the final Dari version was confirmed following a consensus reached during the meeting. After translation, the questions were pretested to identify any items that needed to be better worded and revised. This was done through cognitive interviews using verbal probing as recommended in the literature to assess whether the questions were serving their purposes and to evaluate face validity [10,26].

Two rounds of cognitive interviews were conducted with four Afghan refugee women speaking Dari, and the survey was administered to them at the participants' houses with a female Dari-speaking interpreter, and then the participants were asked their interpretations of the questions, any difficulty they had responding, and any additional situations or circumstances on which their answers were based [27]. All interviews were audio-recorded with consent to inform both the revisions to the scale and practical considerations.

## Results

The initial version of the scale comprises ten domains, including health system knowledge, insurance, making an appointment, transportation, preparing for a visit, in the clinic, interpretation, medicine, medical bills, and preventive care, with a total of 47 items (**S1 Appendix**).

The scale was sent to 21 potentially eligible experts and following an introductory e-mail, 14 of these experts indicated their willingness to participate in the Delphi methods, resulting in a response rate of 66.7%. Fourteen experts completed the survey in Round 1, and 12 completed Round 2. **Table 1** includes the demographic information of the expert panel who participated in the Delphi. The experts represented six states in the US (Arizona, California, Florida, Maryland, Massachusetts, and Pennsylvania). The experts had multiple identities and roles. For instance, several health care providers were involved in refugee health research, and two refugees were health care providers in their home countries. Additionally, a cultural navigator and caseworkers from refugee resettlement agencies had lived experiences as refugees in the past. These diverse backgrounds enabled them to provide more comprehensive insights and a deeper and more nuanced understanding of the scale.

### Round 1

**Table 2** provides an overview of the findings from the Round 1 survey. Among 47 items, a consensus was achieved for 39 items among the expert panel. In accordance with the predefined threshold, eight items with a CVR below 0.62 were eliminated during Round 1. Furthermore, 30 items were modified based on the feedback provided by the expert panel. Some recommendations were made to delete certain items because some services offered to refugees might vary by state. Others suggested improving the wording of certain items, while some

**Table 1. The characteristics of the Delphi panel (N = 14).**

|  |  | n (%) |
|---|---|---|
| **Gender** |  |  |
| Female |  | 12 (85.7) |
| Male |  | 2 (14.3) |
| **Group** |  |  |
|  | Health care provider | 7 (50.0) |
|  | Refugee | 3 (42.9) |
|  | Staff from refugee resettlement agencies | 2 (14.3) |
|  | Cultural navigator | 1 (7.1) |
|  | Researcher | 1 (7.1) |
| **Race/ethnicity** |  |  |
|  | White | 6 (42.9) |
|  | Black or African American | 3 (21.4) |
|  | Asian | 2 (14.3) |
|  | Middle Eastern or North African | 1 (7.1) |
|  | Others (Latino/a) | 2 (14.3) |
|  |  | Mean (range) |
| **Age** |  | 44.6 (33–56) |
| **Years of experience** |  | 13.8 (5–27) |

provided suggestions for concepts such as 'over-the-counter medicine,' which could be challenging for refugees to comprehend.

## Round 2

In Round 2, 12 experts completed the review and only one expert changed the scores after evaluating the revised version. Despite changes in the means as a result of these modifications, there were no changes in the CVR or the exclusion of any items. **Table 3** is the summary from Round 2. Some participants provided feedback to further refine the items. After incorporating feedback from Round 2, the scale was finalized with 10 domains and 39 items and sent for translation into Dari. The finalized version after Round 2 can be found in **S2 Appendix**.

## Translation and cognitive interview and final draft of the HEalthCare NAvigation Competency Scale

After translating the scale, two Afghan refugee women were asked to complete it and share their feedback through a cognitive interview. The feedback encompassed various aspects, such as addressing formatting concerns, enhancing the clarity of translation, considerations for implementation, and addressing other related themes. The suggested changes for translation were discussed with the translators and reflected in the scale. The modified version was tested again with another group of two Afghan refugee women following the same procedure. The feedback and modifications are outlined in **Table 4**, and the revised final scale, reflecting these changes, is available in **S3 Appendix**. Based on the feedback, the scale was finalized with ten domains and 35 items. The factors to consider when implementing the scale, as derived from the scale development process, are described in detail in **Table 5**.

## Discussion

This paper outlined the process of developing and validating the HECNAC Scale for refugees, including content validity assessment through the Delphi, and face validity validation via cognitive interviews. To the best of our knowledge, it is the first endeavor to identify the core

**Table 2. Consensus level among the expert panel on Healthcare Navigation Competency Scale using CVR: Round 1 (N = 14).**

| Domain | Questions | Min. | Max. | Mean | CVR | Consensus Reached |
|---|---|---|---|---|---|---|
| **Healthcare system** (When you have the following conditions or symptoms, what should you do?) | 1. I have difficulty breathing along with chest pain. | 1 | 4 | 1.57 | 0.86 | Yes |
| | 2. I or my children have a mild fever and runny nose. | 1 | 2 | 1.21 | 1.00 | Yes |
| | 3. I think my arm is broken without bleeding or deformation. | 1 | 3 | 1.71 | 0.57 | No |
| | 4. My child needs to be vaccinated. | 1 | 3 | 1.71 | 0.71 | Yes |
| | 5. For the last 6 months, I have experienced stomach pain and constipation. | 1 | 2 | 1.29 | 1 | Yes |
| | 6. I think I (or my wife) got pregnant. | 1 | 2 | 1.36 | 1 | Yes |
| | 7. To see a specialist doctor, I first need to see my primary care doctor (family doctor). | 1 | 4 | 1.64 | 0.57 | No |
| | 8. I have a family doctor. | 1 | 3 | 1.5 | 0.86 | Yes |
| **Insurance** | 9. I can go to the hospital for all health needs at no cost. | 1 | 3 | 1.36 | 0.86 | Yes |
| | 10. I know where to look when I am unsure about which services are and aren't covered by my insurance. | 1 | 3 | 1.5 | 0.86 | Yes |
| | 11. I can get most preventive care (such as immunization, cancer screening) for free with my insurance. | 1 | 3 | 1.29 | 0.86 | Yes |
| **Making an appointment** | 12. I know where to call to make a medical appointment with a family doctor. | 1 | 2 | 1.15 | 1 | Yes |
| | 13. I know how to call and make a medical appointment. | 1 | 2 | 1.23 | 1 | Yes |
| | 14. I have someone who can help me make a medical appointment when needed. | 1 | 3 | 1.31 | 0.69 | Yes |
| **Transportation** | 15. I can go to a clinic either using public transportation or using my car. | 1 | 4 | 1.69 | 0.69 | Yes |
| | 16. I have someone who can give me a ride to a clinic when needed. | 1 | 3 | 1.62 | 0.54 | No |
| | 17. I know what medical taxi is. | 1 | 4 | 2.23 | 0.23 | No |
| | 18. I know how to call and schedule a medical taxi when needed. | 1 | 4 | 2.08 | 0.38 | No |
| **Preparing for a visit** | 19. I know the essential documents to take to a medical appointment. | 1 | 4 | 1.62 | 0.69 | Yes |
| | 20. I usually prepare questions to ask for a doctor's visit. | 1 | 3 | 1.77 | 0.54 | No |
| | 21. When I make an appointment, I ask if there are any dietary recommendations before my appointment such as fasting. | 1 | 4 | 1.77 | 0.69 | Yes |
| | 22.When I make an appointment, I ask if there will be a copayment (money that must be paid by the patient) and how much it will be. | 1 | 3 | 1.46 | 0.69 | Yes |
| | 23. (If I have kids) I have someone to watch my kids during my medical appointment if needed. | 1 | 4 | 1.31 | 0.85 | Yes |
| **In the clinic** | 24. I know how to check in at the reception desk by telling my name or show my ID. | 1 | 2 | 1.23 | 1 | Yes |
| | 25. I know how to fill out necessary paperwork with some help. | 1 | 2 | 1.38 | 1 | Yes |
| | 26. I can express my concerns to my doctor. | 1 | 3 | 1.46 | 0.85 | Yes |
| | 27. I can ask any questions to my doctor. | 1 | 4 | 1.54 | 0.85 | Yes |
| | 28. I know the location of the pharmacy that is close to my house. | 1 | 2 | 1.31 | 1 | Yes |
| | 29. I can let the health care providers know the address of the pharmacy. | 1 | 3 | 1.38 | 0.85 | Yes |
| | 30.At the end of my visit, I ask at the reception desk what to do next (whether I need to return for another visit, need to pick up any medicine, need to go to lab, or get specialist care). | 1 | 3 | 1.46 | 0.69 | Yes |
| | 31. If needed, I know how to get a referral and get specialist care. | 1 | 3 | 1.38 | 0.85 | Yes |
| **Interpretation** | 32. The services of a professional medical interpreter should be provided at no cost to the patients and their family members. | 1 | 3 | 1.69 | 0.54 | No |
| | 33. I know how to request an interpreter at clinic, pharmacy, or over the phone. | 1 | 2 | 1.08 | 1 | Yes |
| **Medicine** | 34. I know the process of getting refills for medicine if necessary. | 1 | 2 | 1.15 | 1 | Yes |
| | 35. I know how to pick up any prescribed or refilled medicines at the pharmacy. | 1 | 1 | 1 | 1 | Yes |
| | 36. I know what the over-the-counter medicine is. | 1 | 2 | 1.38 | 1 | Yes |
| | 37. I know how to get over-the-counter medicine. | 1 | 2 | 1.23 | 1 | Yes |
| | 38. If errors occur at pharmacy—for example, the prescription is not there at the pharmacy, I know what to do and follow up on the issue. | 1 | 3 | 1.31 | 0.85 | Yes |
| | 39. If errors occur at pharmacy—for example, the prescription is not there at the pharmacy, I have someone who can help me solve the issue. | 1 | 3 | 1.46 | 0.69 | Yes |

*(Continued)*

**Table 2.** (Continued)

| Domain | Questions | Min. | Max. | Mean | CVR | Consensus Reached |
|---|---|---|---|---|---|---|
| **Medical bills** | 40. I know how to read the medical bills (either myself or using a translating app such as google translator). | 1 | 3 | 1.54 | 0.85 | Yes |
| | 41. I have someone who can help read the medical bills. | 1 | 3 | 1.31 | 0.85 | Yes |
| | 42. I know how to pay the medical bills if I have money. | 1 | 3 | 1.23 | 0.85 | Yes |
| | 43. If there are any medical billing errors or insurance denies to pay my bills, I know how to address the issues. | 1 | 2 | 1.31 | 1 | Yes |
| | 44. If there are any medical billing errors or insurance denies to pay my bills, I have someone who can help me address the issues. | 1 | 2 | 1.15 | 1 | Yes |
| **Preventive care** | 45. Many illnesses can be prevented through cleanliness, proper nutrition, exercise, and adequate sleep. | 1 | 4 | 1.54 | 0.54 | No |
| | 46. People at certain ages need to get cancer screening even though they don't feel sick or don't have any symptoms. | 1 | 3 | 1.31 | 0.85 | Yes |
| | 47. Vaccinations are effective in preventing some diseases. | 1 | 2 | 1.08 | 1 | Yes |

Abbreviations: CVR, Content validity ratio (between 1 and -1 with the higher score indicating further agreement among the panel), a CVR below the predetermined cutoff point of 0.62 was eliminated.

Responses for the item 1–6 were: Treat at home, go to a primary (family) doctor, go to an urgent care, go to emergency care, call 911, unsure/don't know. Responses for the remaining items were: Strongly disagree, disagree, neither agree or disagree, agree, strongly agree.

Each item was assessed by the expert panel using a 4-point scale, with a score of 1 indicating high relevance and a score of 4 indicating irrelevance.

competencies required to navigate the US healthcare system, particularly for refugee communities in the country. Rather than simply focusing on health literacy, which often assumes proficiency in English, the HECNAC Scale captures multidimensional facets of healthcare navigation such as social support. For example, even though a refugee is not able to schedule a medical appointment by him/herself, having someone who can assist with the process by arranging appointments is considered as an advantage compared to those lacking such support. The scale also considers other barriers that often pose challenges to healthcare access such as language barriers or transportation. However, one limitation of this study is its exclusive focus on assessing content validity and face validity, which represent only some aspects of scale evaluation. Therefore, future research may be valuable to examine other dimensions of the scale, such as reliability and criterion validity, through factor analysis [10]. Additionally, the face validity was examined using a single group of refugee communities, specifically Dari-speaking Afghans. Despite the growing number of Afghan refugees since the Taliban's control of Afghanistan in 2021, they represent only a small segment of the overall refugee population in the US, which is highly diverse with distinct needs and cultures. Therefore, it may be beneficial to conduct similar tests with different refugee groups to validate the scale's applicability.

Overall, this study is a significant step towards understanding the core competencies of refugees when navigating the complex healthcare system in the US and the ways in which the competencies can be measured. Once the scale is validated against other properties in terms of scale evaluation, it has the potential to be adaptable and scalable to other immigrant groups in the US, as they share similar challenges in navigating unfamiliar healthcare systems.

## Conclusions

Programs designed to support refugees in the US tend to be short-term, and numerous research studies have indicated that refugees often encounter difficulties in navigating the healthcare system in the country, even after many years of resettlement. As a result, it is

**Table 3. Consensus level among the expert panel on Healthcare Navigation Competency Scale using CVR: Round 2 (N = 12).**

| Domain | Questions | Min. | Max. | Mean | CVR | Consensus Reached |
|---|---|---|---|---|---|---|
| Healthcare system (When you have the following conditions or symptoms, what should you do?) | 1. I have difficulty breathing along with chest pain. | 1 | 4 | 1.57 | 0.86 | Yes |
| | 2. I have a mild fever (below 103F) and runny nose. | 1 | 2 | 1.21 | 1.00 | Yes |
| | 3. I think my arm is broken without bleeding or deformation. | 1 | 3 | 1.71 | 0.57 | No |
| | 4. I need to be vaccinated. | 1 | 3 | 1.71 | 0.71 | Yes |
| | 5. For the last 6 months, I have occasional experienced stomach pain and constipation. | 1 | 2 | 1.21 | 1 | Yes |
| | 6. I think I (or my wife) got pregnant. | 1 | 2 | 1.36 | 1 | Yes |
| | 7. To see a specialist doctor, I first need to see my primary care doctor (family doctor). | 1 | 4 | 1.64 | 0.57 | No |
| | 8. I need to have a family doctor or primary care provider. | 1 | 3 | 1.5 | 0.86 | Yes |
| Insurance | 9. I can go to the hospital for all health needs at no cost. | 1 | 3 | 1.36 | 0.86 | Yes |
| | 10. I know where to learn more when I am unsure whether a treatment is covered by my insurance. | 1 | 3 | 1.5 | 0.86 | Yes |
| | 11. I can get most preventive care (such as immunization, cancer screening) for free with my insurance. | 1 | 3 | 1.29 | 0.86 | Yes |
| Making an appointment | 12. I know where to call to make a medical appointment with a family doctor or primary care provider. | 1 | 2 | 1.15 | 1 | Yes |
| | 13. I am able to call and make a medical appointment by myself. | 1 | 2 | 1.15 | 1 | Yes |
| | 14. I have someone who can help me make a medical appointment when needed. | 1 | 3 | 1.31 | 0.69 | Yes |
| Transportation | 15. I have access to transport to get to my medical appointment. | 1 | 4 | 1.69 | 0.69 | Yes |
| | 16. I have someone who can give me a ride to a clinic when needed. | 1 | 3 | 1.62 | 0.54 | No |
| | 17. I know what medical taxi is. | 1 | 4 | 2.23 | 0.23 | No |
| | 18. I know how to call and schedule a medical taxi when needed. | 1 | 4 | 2.08 | 0.38 | No |
| Preparing for a visit | 19. I know the essential documents to take to a medical appointment. | 1 | 4 | 1.62 | 0.69 | Yes |
| | 20. I usually prepare questions to ask for a doctor's visit. | 1 | 3 | 1.77 | 0.54 | No |
| | 21. When I make an appointment, I ask if there are any dietary recommendations before my appointment such as fasting. | 1 | 4 | 1.77 | 0.69 | Yes |
| | 22. When I make an appointment, I ask if there will be a copayment (money that must be paid by the patient) and how much it will be. | 1 | 3 | 1.46 | 0.69 | Yes |
| | 23. (If I have kids) I have someone to watch my kids during my medical appointment if needed. (not applicable) | 1 | 4 | 1.31 | 0.85 | Yes |
| In the clinic | 24. I know how to check in at the reception desk by telling my name and date of birth or show my ID. | 1 | 2 | 1.15 | 1 | Yes |
| | 25. I am able to fill out necessary paperwork by myself or I have someone who can help the process. | 1 | 2 | 1.31 | 1 | Yes |
| | 26. I feel comfortable discussing my concerns with my health care provider. | 1 | 3 | 1.46 | 0.85 | Yes |
| | 27. I feel comfortable asking any questions to my health care provider. | 1 | 4 | 1.54 | 0.85 | Yes |
| | 28. I know where to go when a prescription is ordered. | 1 | 2 | 1.31 | 1 | Yes |
| | 29. I am able to let the health care providers know my preferred pharmacy either by telling them or showring my ID. | 1 | 3 | 1.38 | 0.85 | Yes |
| | 30. I know what to expect after my visit and when I should return if necessary. | 1 | 3 | 1.46 | 0.69 | Yes |
| | 31. If needed, I know how to get specialist care. | 1 | 3 | 1.38 | 0.85 | Yes |
| Interpretation | 32. The services of a professional medical interpreter should be provided at no cost to the patients and their family members. (I have the right to an interpreter at any medical visit at no cost.) | 1 | 3 | 1.62 | 0.54 | No |
| | 33. I know how to request an interpreter at clinic, pharmacy, or over the phone. | 1 | 2 | 1.08 | 1 | Yes |

*(Continued)*

**Table 3.** (Continued)

| Domain | Questions | Min. | Max. | Mean | CVR | Consensus Reached |
|---|---|---|---|---|---|---|
| **Medicine** | 34. I am able to ask for more medication from my doctor if necessary. | 1 | 2 | 1.15 | 1 | Yes |
| | 35. I am able to pick up any prescribed or refilled medicines at the pharmacy. | 1 | 1 | 1 | 1 | Yes |
| | 36. I am aware of medications that do not require a prescription from a health care provider. | 1 | 2 | 1.38 | 1 | Yes |
| | 37. I am able to get medication that does not need a prescription from a pharmacy. | 1 | 2 | 1.23 | 1 | Yes |
| | 38. I know what to do when I cannot get my prescription on time (for example, a prescription is not there, something is wrong with medication). | 1 | 3 | 1.31 | 0.85 | Yes |
| | 39. I have someone who can help me when I cannot get my prescription on time. | 1 | 3 | 1.46 | 0.69 | Yes |
| **Medical bills** | 40. I am able to understand medical bills (either myself or using a translating app). | 1 | 3 | 1.54 | 0.85 | Yes |
| | 41. I have someone who can help understand medical bills. | 1 | 3 | 1.31 | 0.85 | Yes |
| | 42. I know how to pay the medical bills when I need to. | 1 | 3 | 1.23 | 0.85 | Yes |
| | 43. If there are any medical billing errors or insurance denies paying my bills, I know how to address the issues. | 1 | 2 | 1.31 | 1 | Yes |
| | 44. If there are any medical billing errors or insurance denies paying my bills, I have someone who can help me address the issues. | 1 | 2 | 1.15 | 1 | Yes |
| **Preventive care** | 45. Many illnesses can be prevented through cleanliness, proper nutrition, exercise, and adequate sleep. | 1 | 4 | 1.54 | 0.54 | No |
| | 46. People at certain ages need to get certain tests to check their bodies for possible illness like cancer, even if they don't feel sick. | 1 | 3 | 1.23 | 0.85 | Yes |
| | 47. Vaccinations are effective in preventing some diseases. | 1 | 2 | 1.08 | 1 | Yes |

Abbreviations: CVR, Content validity ratio (between 1 and -1 with the higher score indicating further agreement among the panel), a CVR below the predetermined cutoff point of 0.62 was eliminated.

Responses for the item 1–6 were: Treat at home, go to a primary (family) doctor, go to an urgent care, go to emergency care, call 911, unsure/don't know. Responses for the remaining items were: Strongly disagree, disagree, neither agree or disagree, agree, strongly agree.

Each item was assessed by the expert panel using a 4-point scale, with a score of 1 indicating high relevance and a score of 4 indicating irrelevance.

**Table 4. Changes made to the scale based on the cognitive interviews.**

| Domain/ Item | Feedback | Changes made to the items |
|---|---|---|
| **Health System Knowledge** | As the instruction for the Health System Knowledge domain is embedded within the table, it was not easily distinguishable. | To improve clarity, the instruction was removed from the table and presented before the table with a more detailed explanation. |
| **Health System Knowledge** | While the questions are posed as hypothetical scenarios, refugees, particularly those with limited education, struggled to understand these hypothetical questions and found it challenging to respond if they lacked relevant experiences (For instance, when the interviewee is engaged in family planning and has no intentions of having additional children, they were uncertain about which answer to select for item 6.). | Changed the sentences from a first-person perspective to scenarios involving another person and inquired about what actions that person should take to emphasize that the situation is hypothetical. |
| **Health System Knowledge Item 2** | Refugees may originate from countries using different measurement systems such as Celsius rather than Fahrenheit | Both scales were added for better clarity. |
| **Making an appointment Items 11, 12 Medicine Items 31, 32 Medical bills Items 33, 34, 36, 37** | Some questions were divided intentionally to determine whether refugees possess the competency to perform certain tasks independently or if they have someone for assistance, thus gauging their level of social support. However, it was observed that when refugees are capable of completing these tasks on their own, they tended to mark the following questions as 'strongly disagree' or 'disagree' to convey that they do not require external support. | What is critical in this scale is whether they possess the competency to independently perform tasks OR if they have someone to rely on for assistance to identify the most vulnerable refugees, those who lack both the competency and a support system, the two items are combined. |

**Table 5. Practical considerations when implementing the scale.**

• Refugees particularly female refugees might prefer meeting at their homes due to childcare or transportation challenges.
• Having a facilitator proficient in both English and the respondent's native language is essential, regardless of the respondent's English proficiency. Even if a refugee is proficient in their own language and can read and write, they may still encounter difficulties understanding foreign concepts like over-the-counter medicine and specialist care.
• The facilitator must be well-trained and knowledgeable about the scale's questions to address any queries that may arise during its completion.
• In addition to the scale, collecting demographic information, such as the length of stay in the US, ongoing assistance from a refugee resettlement agency, and support from Medicaid, can provide a more nuanced understanding of the respondent's context.
• During the scale implementation, other needs may emerge. Depending on the scale's purpose, it may be advantageous to thoroughly inquire about them and document these needs alongside the quantitative responses.

crucial to identify the competencies necessary to effectively navigate the healthcare system, particularly for refugee communities. This HECNAC Scale can be used as a tool to assess levels of competency among refugees in this regard and identify the gaps and challenges they face. In doing so, it could contribute to providing more tailored interventions to refugees with varying levels of competency and connecting those with limited competency with community resources. Additionally, based on these identified competencies, training curricula can also be standardized to ensure consistency and effectiveness across different agencies and caseworkers in the country. The HECNAC Scale could serve as the first step in the journey towards mitigating the persistent barriers refugees continue to face, even long after resettling in the country.

## Supporting information

**S1 Appendix. The first draft of the Healthcare Navigation Competency Scale.**
(DOCX)

**S2 Appendix. The second version of the Healthcare Navigation Competency Scale after the Delphi.**
(DOCX)

**S3 Appendix. The final version of the Healthcare Navigation Competency Scale.**
(DOCX)

## Acknowledgments

We express our gratitude to Dr. Mike Edwards for his insightful comments and feedback on this study.

## Author Contributions

**Conceptualization:** Sarah Yeo, Inseok Lee, Priscilla Magrath, Kacey Ernst, Halimatou Alaofè.

**Formal analysis:** Sarah Yeo.

**Funding acquisition:** Sarah Yeo.

**Investigation:** Sarah Yeo.

**Methodology:** Sarah Yeo, Inseok Lee.

**Project administration:** Sarah Yeo.

**Resources:** Sarah Yeo, John Ehiri.

**Software:** Sarah Yeo.

**Supervision:** John Ehiri, Priscilla Magrath, Kacey Ernst, Halimatou Alaofè.

**Validation:** Sarah Yeo.

**Writing – original draft:** Sarah Yeo.

**Writing – review & editing:** Sarah Yeo, Inseok Lee, John Ehiri, Priscilla Magrath, Kacey Ernst, Yu Ri Kim, Halimatou Alaofè.

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
