## [Decision Letter · Decision Letter 0]

1 Jul 2024

PONE-D-24-08489Developing and validating a HEalthCare NAvigation Competency (HECNAC) Scale for refugees in the United StatesPLOS ONE

Dear Dr. Yeo,

Thank you for submitting your manuscript to PLOS ONE. After careful consideration, we feel that it has merit but does not fully meet PLOS ONE’s publication criteria as it currently stands. Therefore, we invite you to submit a revised version of the manuscript that addresses the points raised during the review process.

This study appears to be a good addition to the current literature. It addresses health care needs and understanding among refugees from an ethnic population for which little information is currently available. There are some areas in the manuscript needing improvement. In particular, key details such as target population and research approach are sketchy in the background. That section needs more refinement. In addition, the methods and results section are overlapping; methods details should be all appear in that section and moved from the findings. Further clarifications and language edits are also needed throughout the manuscript. All comments from both reviewers should be addressed carefully in the revision.

We look forward to receiving your revised manuscript.

Kind regards,

Magdalena Szaflarski, PhD

Academic Editor

PLOS ONE

“This research is supported in part by NIH T32 CA078447.”

3. In the online submission form, you indicated that [The data underlying the results presented in the study are available upon request].

Reviewers' comments:

Reviewer's Responses to Questions

**Comments to the Author**

1. Is the manuscript technically sound, and do the data support the conclusions?

Reviewer #1: Yes

Reviewer #2: Yes

2. Has the statistical analysis been performed appropriately and rigorously? 

Reviewer #1: N/A

Reviewer #2: Yes

3. Have the authors made all data underlying the findings in their manuscript fully available?

Reviewer #1: Yes

Reviewer #2: Yes

4. Is the manuscript presented in an intelligible fashion and written in standard English?

Reviewer #1: Yes

Reviewer #2: Yes

5. Review Comments to the Author

Reviewer #1: Thank you for the opportunity to invite me reviewing this paper titled” Developing and validating a HEalthCare NAvigation Competency (HECNAC) Scale for refugees in the United States”. It is a very interesting topic and the HECNAC scale might have an important role in supporting with healthcare navigation and tailored interventions as well as improve accessibility in refugees’ populations in US, and perhaps can be adapted in many other Western countries for other migrants’ population. Please see the detailed comments in the attached file.

Reviewer #2: Thank you for this very interesting study - the results will be very useful and immediately applicable in the field. There are some areas in the text that aren't well-communicated, but with some clarification this will be a very strong study.

I've included here a mixture of minor and major comments:

1. Line 38, page 12: Think you mean, Therefore? Subsequently is awkward here

2. Figure 1 - 2 things: would say 1st round and 2nd round... also clarify whether 2nd set of 39 is same or different

3. Page 15, Line 110 - numerous variations "of the" Delphi would flow better

4. Page 16, Line 115 - What is the geographic distribution of your group? healthcare challenges in the US have regional characteristics as well as universal ones. Also the refugee populations are different and thus the systems they are used to vary widely and staff have to know those groups, so across the country would have different experiences. would be good to comment on this (I see that you report this later on but would be good to discuss in the discussion)

5. Page 16, Line 123 - The sentence starting with Niederberger and Spranger is confusing... meaning on average the number of experts is low to medium double digit? Saying 'the Delphi studies' sounds like you are referring to specific ones... is that what you mean? If so, which ones?

6. In the same paragraph as the sentence above a minor note, but if you use the name for one study, would do it for all (or do it for none) unless Niederberger and Spranger are well-known as originators of the method or some other founding role.

7. Page 17, Line 151 - but only 14 accepted? The others refused?

8. Section on translation - you need to introduce the focus on Afghan refugees MUCH earlier that this is the language of focus and WHY

9. Page 18, Line 176 cognitive interviews with whom? how were they identified?

10. Page 19, Line 179 this needs to come at the beginning of the section

11. Page 19, Line 183 and transcribed? what did you do with the recordings?

12. The methods and results seem mixed together.... a lot of the detail I felt was missing in methods appears here in results. If you want to organize it that way, then I would keep the methods much more high level and use the results to provide the detail. OR put all of this detail in the methods and really focus the results on describing what came out of the whole process vs. the process. I'd definitely suggest modeling your set up on other Delphi papers... how do they approach this?

13. Page 19, Line 195 Some of these really seem more like roles rather than identities.

14. Table 1 - spacing in 'Group' is off; Race/ethnicity: Why group the others if you have given specificity for the one person who is Middle Eastern, there is no reason for an 'other' category unless they refused

15. Page 21, Line 232 This (note about table 5) needs more context... did this come from the Delphi discussions?? the table below felt like a surprise despite this... the suggestions are excellent, they just need more lead in; Table 5: This needs prep in the results... it kind of comes out of nowhere here in the flow.

16. page 23, Line 255 yes re: the limitation of focusing on Afghans - and this needs much better intro early in the paper

17. same page, line 260 as you have elsewhere I'd say refugees and immigrants

6. PLOS authors have the option to publish the peer review history of their article (what does this mean?). If published, this will include your full peer review and any attached files.

Reviewer #1: **Yes: **Wendan Shi

Reviewer #2: No

---

## [Author Response · Author response to Decision Letter 0]

3 Oct 2024

Dear Editor and Reviewers for PLOS ONE, 

Thank you so much for providing the opportunity to submit this revised manuscript titled “Developing and validating a HEalthCare NAvigation Competency (HECNAC) Scale for refugees in the United States” to PLOS ONE. 

We appreciate your insightful and valuable feedback on our manuscript. It substantially helped enrich the manuscript overall. We have incorporated your suggestions accordingly and highlighted changes within the manuscript using the track-change function. Our responses to each point are below in blue. If you have any additional suggestions or concerns, please kindly let us know.

Reviewer #1: Thank you for the opportunity to invite me reviewing this paper titled” Developing and validating a HEalthCare NAvigation Competency (HECNAC) Scale for refugees in the United States”. It is a very interesting topic and the HECNAC scale might have an important role in supporting with healthcare navigation and tailored interventions as well as improve accessibility in refugees’ populations in US, and perhaps can be adapted in many other Western countries for other migrants’ population. Please see the detailed comments in the attached file.

Below is from the attached file. 

Thank you for the opportunity to invite me reviewing this paper titled” Developing and validating a HEalthCare NAvigation Competency (HECNAC) Scale for refugees in the United States”. It is a very interesting topic and the HECNAC scale might have an important role in supporting with healthcare navigation and tailored interventions as well as improve accessibility in refugees’ populations in US, and perhaps can be adapted in many other Western countries for other migrants’ population. Here is a list of detailed comments:

Abstract: Too many background information, and more methods and results are needed such as the description of panels’ characteristics (e.g., number of panels, professions etc.). Also, domains of the scale can be examined in full.

Response: Thank you so much for sharing your insight. As you suggested, we reduced some background information. Instead, more methodological components and result sections were included. We included the domains of the scale as per your suggestion within the word limit of the abstract. 

Introduction: It will be interesting to read more about refugees in US, the history, ethnicity, and common languages they spoke at home. Some statistics about this population proportion and background can help to get more understand of the need in developing this scale.

Response: This is an excellent point. We incorporated key statistics in lines 15-16. 

Methods: Is there any other healthcare navigation scale available? It is not clear in Line 74-77 about what frameworks, models and scales available. Please clarify.

Response: Currently available tools to measure navigation competency were discussed extensively in lines 42-60. 

Figure 1: please list the stakeholders in the box, e.g., how many refugees, healthcare providers etc. How did those stakeholders been selected? (12 females and 2 males, 6 white) Please clarify.

Response: Great point. We added the number of stakeholders in each category and further details concerning the inclusion criteria and recruitment process (lines 87-93). In addition, we included the reference for the paper that detailed the methodology and processes (lines 98-99).

Results: How long does participant take to complete the scale? Despite the language, are there any feedback regarding the cultural difference and barriers? Please expand more. Will the cognitive interview guide/questions be available? How long are the interviews?

Response: If this question is related to the cognitive interviews, we believe it is a challenging question to answer as some participants completed the scale on their own while others received assistance from the interpreter. Feedback from the respondents and changes made to the items are detailed in Table 4. We did not include the cognitive interview guide/questions in full as we followed the protocols and examples illustrated in the references (25 and 26). The questions are also summarized in lines 206-211. 

I really like Table 5 the practical considerations. Are they direct quotation from the interviewees? Or has been paraphrased by researcher? If they are quotes, it might be worth to add which ones from the consumers (refuges) and which ones from the healthcare providers. If they have been summarised, please make them more precise, such as the second point is very lengthy.

Response: These were practical considerations derived from the entire scale development process, facilitating the scale through cognitive interviews and interacting and observing interpreters through the interviews. We made this clearer by modifying the relevant section (line 268) as per your feedback. Concerning your second point, we divided the points to mitigate your concern (table 5). 

Discussion and conclusion: this paper is focused on one refugee group who are Dari-speaking Afghans. However, the language Dari is first appeared in mid-Methods, and the Afghan refugee group is not stated elsewhere in the Intro, aim or method sections. Please introduce this population earlier in the paper for clarification.

Response: Thank you for highlighting this important point. Based on your point, these factors that influenced the decision were added in lines 187-192. 

(We did not include a discussion of Dari-speaking Afghans in the introduction or research aims because the scale is not solely intended for Afghan refugees. However, due to time and resource limitations, it was impractical to test the scale across all ethnic groups from different countries in the United States. Consequently, we decided to begin by testing the scale with a specific population. And this decision was based on two key factors. Firstly, the need arose due to a significant increase in Afghan refugees in the US after the Taliban’s takeover in 2019, as community partners highlighted the necessity of aiding their navigation of the healthcare system. Secondly, the first author’s previous work and established connections with the population provided easy access to them.) 

Reviewer #2: Thank you for this very interesting study - the results will be very useful and immediately applicable in the field. There are some areas in the text that aren't well-communicated, but with some clarification this will be a very strong study.

I've included here a mixture of minor and major comments:

1. Line 38, page 12: Think you mean, Therefore? Subsequently is awkward here

Response: Thank you for your suggestion. We made a change accordingly (line 40). 

2. Figure 1 - 2 things: would say 1st round and 2nd round... also clarify whether 2nd set of 39 is same or different

Response: That is something we missed. Thank you for bringing this up. We incorporated your suggestion and modified the figure to make it clearer. 

3. Page 15, Line 110 - numerous variations "of the" Delphi would flow better

Response: We made the change based on your suggestion. 

4. Page 16, Line 115 - What is the geographic distribution of your group? healthcare challenges in the US have regional characteristics as well as universal ones. Also the refugee populations are different and thus the systems they are used to vary widely and staff have to know those groups, so across the country would have different experiences. would be good to comment on this (I see that you report this later on but would be good to discuss in the discussion)

Response: That is such an excellent point. Indeed, each US state has different laws, regulations, different entitlement for refugees and this may lead to diverse experience among refugees even from the same countries. We also noticed that even in the same state, depending on the refugee resettlement agencies and their capacity and level of experience and knowledge of caseworkers, the experiences of refugees tend to vary. Acknowledging the regional differences, we tried to recruit experts from different regions in the US as noted in line 219. Although we were not able to recruit experts from all the regions, we would say that this expert panel from 6 states could speak to different experiences of refugees. 

5. Page 16, Line 123 - The sentence starting with Niederberger and Spranger is confusing... meaning on average the number of experts is low to medium double digit? Saying 'the Delphi studies' sounds like you are referring to specific ones... is that what you mean? If so, which ones?

Response: Yes, that is correct. The phrase was taken from the article and we used quotation marks to make it clearer. (“The average number of experts included was usually in the low to medium double-digit range (e.g., ID1: median = 17 invited experts; ID11: mean = 40 experts in the first Delphi round).”) And we were not referring to specific studies and we took out “the” (lines 137-140). 

6. In the same paragraph as the sentence above a minor note, but if you use the name for one study, would do it for all (or do it for none) unless Niederberger and Spranger are well-known as originators of the method or some other founding role.

Response: Thank you so much for your suggestion. 

7. Page 17, Line 151 - but only 14 accepted? The others refused?

Response: Yes, as noted in line 148, 21 experts were reached and 14 experts participated in the Delphi. More information can be found in lines 221-223. As a recommended practice, we included the response rate as well (line 222). 

8. Section on translation - you need to introduce the focus on Afghan refugees MUCH earlier that this is the language of focus and WHY

Response: Great point. Based on your suggestion, we included another paragraph earlier in the manuscript and you can find it in lines 187-192. 

9. Page 18, Line 176 cognitive interviews with whom? how were they identified?

Response: The information can be found in lines 206-211. 

10. Page 19, Line 179 this needs to come at the beginning of the section

Response: The translation was conducted prior to the cognitive interviews and that part was described chronologically. 

11. Page 19, Line 183 and transcribed? what did you do with the recordings?

Response: It was used to inform the necessary changes to be made to the scale and practical considerations. It was added to the manuscript to address your suggestion (line 211). 

12. The methods and results seem mixed together.... a lot of the detail I felt was missing in methods appears here in results. If you want to organize it that way, then I would keep the methods much more high level and use the results to provide the detail. OR put all of this detail in the methods and really focus the results on describing what came out of the whole process vs. the process. I'd definitely suggest modeling your set up on other Delphi papers... how do they approach this?

Response: The first part of the results is a description of the demographic information of the expert panel, which is common for the first part of a result section. The second part was describing the results from Round 1 and Round 2 and subsequent changes as a result of the suggestions from the panel. The last part is the results from the cognitive interviews. Is there a specific part that you believe a better fit for the methods section? We would like to address this concern if you could specify the section. 

13. Page 19, Line 195 Some of these really seem more like roles rather than identities.

Response: We changed the wording to “identities and roles” as it also includes identity as refugees in line 226-227 (“a cultural navigator and caseworkers from refugee resettlement agencies had lived experiences as refugees in the past”)

14. Table 1 - spacing in 'Group' is off; Race/ethnicity: Why group the others if you have given specificity for the one person who is Middle Eastern, there is no reason for an 'other' category unless they refused

Response: There was someone who did not specify their race/ethnicity, and we had to have the others category. 

15. Page 21, Line 232 This (note about table 5) needs more context... did this come from the Delphi discussions?? the table below felt like a surprise despite this... the suggestions are excellent, they just need more lead in; Table 5: This needs prep in the results... it kind of comes out of nowhere here in the flow.

Response: Thank you for your suggestion. More contextual information is added in line 268. It was tied to your previous suggestions concerning the recording of cognitive interviews. We hope the responses to your feedback 11 also clarifies this issue. 

16. page 23, Line 255 yes re: the limitation of focusing on Afghans - and this needs much better intro early in the paper

Response: As per your suggestion, this part was added in lines 186-191. 

17. same page, line 260 as you have elsewhere I'd say refugees and immigrants

Response: It is saying that it has the potential to be scalable to immigrant populations OTHER THAN refugees. So in this context, we would say immigrants rather than refugees and immigrants. We revised this to make it clear in lines 306-307. 

Thank you so much for your valuable feedback. If there are additional feedback or any other lingering concerns, we are happy to address them.

---

## [Decision Letter · Decision Letter 1]

29 Oct 2024

PONE-D-24-08489R1Developing and validating a HEalthCare NAvigation Competency (HECNAC) Scale for refugees in the United StatesPLOS ONE

Dear Dr. Yeo,

Thank you for submitting your manuscript to PLOS ONE. After careful consideration, we feel that it has merit but does not fully meet PLOS ONE’s publication criteria as it currently stands. Therefore, we invite you to submit a revised version of the manuscript that addresses the points raised during the review process. The revised manuscript reads much better and clarifies previous questions/comments about the study. However, minor further revisions are still recommended. Specifically, the context of focusing on Dari speaking Afghans needs much earlier introduction as a key contextual factor, and the study the limitations need more refinement. Please see Comments from Reviewer 2 for further details.

We look forward to receiving your revised manuscript.

Kind regards,

Magdalena Szaflarski, PhD

Academic Editor

PLOS ONE

Journal Requirements:

Reviewers' comments:

Reviewer's Responses to Questions

**Comments to the Author**

1. If the authors have adequately addressed your comments raised in a previous round of review and you feel that this manuscript is now acceptable for publication, you may indicate that here to bypass the “Comments to the Author” section, enter your conflict of interest statement in the “Confidential to Editor” section, and submit your "Accept" recommendation.

Reviewer #1: All comments have been addressed

Reviewer #2: (No Response)

2. Is the manuscript technically sound, and do the data support the conclusions?

Reviewer #1: Yes

Reviewer #2: Yes

3. Has the statistical analysis been performed appropriately and rigorously? 

Reviewer #1: Yes

Reviewer #2: Yes

4. Have the authors made all data underlying the findings in their manuscript fully available?

Reviewer #1: Yes

Reviewer #2: Yes

5. Is the manuscript presented in an intelligible fashion and written in standard English?

Reviewer #1: Yes

Reviewer #2: Yes

6. Review Comments to the Author

Reviewer #1: Thank you for addressing those comments and the revised manuscript looks great. No further comments. Thank you.

Reviewer #2: I appreciate the edits the authors have made to the manuscript. It reads much more clearly now and will be a valuable contribution. I appreciate the context added regarding the influx of Afghan refugees since 2019 as well as the explanation of the outreach in the lead investigator's circle. That said, I still think the context of focusing on Dari speaking Afghans needs much earlier introduction - e.g. in the intro - as a contextual factor, not to minimize the value of the scale developed here nor the importance of this group - likely an understudied group!, but rather because the scale definitively DOES need testing in other refugee groups to ensure the assumptions hold up. On that point, the limitations need to be strengthened, too.

7. PLOS authors have the option to publish the peer review history of their article (what does this mean?). If published, this will include your full peer review and any attached files.

Reviewer #1: **Yes: **Wendan Shi

Reviewer #2: No

---

## [Author Response · Author response to Decision Letter 1]

1 Nov 2024

Reviewer #2: I appreciate the edits the authors have made to the manuscript. It reads much more clearly now and will be a valuable contribution. I appreciate the context added regarding the influx of Afghan refugees since 2019 as well as the explanation of the outreach in the lead investigator's circle. That said, I still think the context of focusing on Dari speaking Afghans needs much earlier introduction - e.g. in the intro - as a contextual factor, not to minimize the value of the scale developed here nor the importance of this group - likely an understudied group!, but rather because the scale definitively DOES need testing in other refugee groups to ensure the assumptions hold up. On that point, the limitations need to be strengthened, too.

Response: Thank you for your suggestion. We moved the section which details the rationale for choosing the Afghan population for testing this scale in the introduction section (lines 63-69). We also strengthened our limitation section based on your suggestion (lines 285-289).

---

## [Editor Report · Decision Letter 2]

5 Nov 2024

Developing and validating a HEalthCare NAvigation Competency (HECNAC) Scale for refugees in the United States

PONE-D-24-08489R2

Dear Dr. Yeo,

We’re pleased to inform you that your manuscript has been judged scientifically suitable for publication and will be formally accepted for publication once it meets all outstanding technical requirements.

Kind regards,

Magdalena Szaflarski, PhD

Academic Editor

PLOS ONE
---

## [Editor Report · Acceptance letter]

16 Dec 2024

PONE-D-24-08489R2 

PLOS ONE

Dear Dr. Yeo, 

I'm pleased to inform you that your manuscript has been deemed suitable for publication in PLOS ONE. Congratulations! Your manuscript is now being handed over to our production team.

Kind regards, 

on behalf of

Dr. Magdalena Szaflarski 

Academic Editor

PLOS ONE